# Deep learning models predict regulatory variants in pancreatic islets and refine type 2 diabetes association signals

**Agata Wesolowska-Andersen[1], Grace Zhuo Yu[2], Vibe Nylander[2], Fernando Abaitua[1], Matthias Thurner[1,2], Jason M Torres[1], Anubha Mahajan[1], Anna L Gloyn[1,2,3†], Mark I McCarthy[1,2,3†‡*]**

[1]Wellcome Centre for Human Genetics, Oxford, United Kingdom; [2]Oxford Centre for Diabetes, Endocrinology and Metabolism, University of Oxford, Oxford, United Kingdom; [3]Oxford NIHR Biomedical Centre, Churchill Hospital, Oxford, United Kingdom

**Abstract** Genome-wide association analyses have uncovered multiple genomic regions associated with T2D, but identification of the causal variants at these remains a challenge. There is growing interest in the potential of deep learning models - which predict epigenome features from DNA sequence - to support inference concerning the regulatory effects of disease-associated variants. Here, we evaluate the advantages of training convolutional neural network (CNN) models on a broad set of epigenomic features collected in a single disease-relevant tissue – pancreatic islets in the case of type 2 diabetes (T2D) - as opposed to models trained on multiple human tissues. We report convergence of CNN-based metrics of regulatory function with conventional approaches to variant prioritization – genetic fine-mapping and regulatory annotation enrichment. We demonstrate that CNN-based analyses can refine association signals at T2D-associated loci and provide experimental validation for one such signal. We anticipate that these approaches will become routine in downstream analyses of GWAS.

**\*For correspondence:**
mark.mccarthy@drl.ox.ac.uk

†These authors contributed equally to this work

**Present address:** ‡Genentech, South San Francisco, United States

## Introduction

Genome-wide association studies (GWAS) have identified over 400 independent signals implicated in genetic susceptibility to type 2 diabetes (T2D) (*Mahajan et al., 2018*). However, efforts to derive biological insights from these signals face the challenge of identifying the functional, causal variants driving these associations within the sets of credible variants defined by the linkage disequilibrium (LD) structure at each locus. It remains far from trivial to assign mechanisms of action at these loci: most associated variants map to non-coding sequence, the implication being that they influence disease risk through the transcriptional regulation of one or more of the nearby genes.

There is mounting evidence that disease-associated variants are likely to perturb genes and regulatory modules that are of specific importance within disease-relevant cell types or tissues (*Marbach et al., 2016*; *Battle et al., 2017*). For example, several studies have reported significant enrichment of T2D GWAS variants within pancreatic islet enhancer regions (*Parker et al., 2013*; *Pasquali et al., 2014*), with that enrichment particularly concentrated in subsets of islet enhancers characterised by open chromatin and hypomethylation (*Thurner et al., 2018*), and clustered in 3D enhancer hub structures (*Miguel-Escalada et al., 2018*). Pancreatic islets represent a key tissue for the maintenance of normal glucose homeostasis, and uncovering islet-specific regulatory mechanisms is therefore critical to understanding T2D aetiology and pathogenesis.

Several studies have generated genome-wide epigenomic profiling datasets of whole human pancreatic islets, and/or FACS-sorted individual islet cell types (*Bhandare et al., 2010*; *Gaulton et al.,*

*2010*; *Stitzel et al., 2010*; *Parker et al., 2013*; *Pasquali et al., 2014*; *Maher, 2012*; *Thurner et al., 2018*; *Bramswig et al., 2013*; *Ackermann et al., 2016*). This wealth of genomic data provides a valuable resource for studying the regulatory machinery of human pancreatic islets, and has proven instrumental in prioritising disease-associated variants by considering their overlap with regulatory elements enriched in disease-associated signals (*Huang et al., 2017*; *Thurner et al., 2018*). However, in cases where multiple associated variants in high LD reside in the same regulatory region, a method with higher resolution is needed to resolve the causal variant. High-throughput massively parallel reporter assays (MPRA) offer one solution for the empirical assessment of putatively functional variants (*Ulirsch et al., 2016*; *Tewhey et al., 2016*), but they are expensive to deploy genome-wide, and may not fully recapitulate the cellular context.

Convolutional neural networks (CNNs) are emerging as a powerful tool to study regulatory motifs in genomic data, and are well suited to extracting high-level information from high-throughput datasets de novo. Indeed, CNN frameworks have been shown to aid in prioritization of genomic variants based on their predicted effect on chromatin accessibility and modifications (*Zhou and Troyanskaya, 2015*; *Kelley et al., 2016*) or gene expression (*Zhou et al., 2018*; *Kelley et al., 2018*). The methods deployed so far learn the regulatory code de novo from genomic sequences of regulatory regions gathered from multiple tissues, using datasets provided by the ENCODE (*Maher, 2012*), the NIH Epigenome Roadmap (*Bernstein et al., 2010*) and GTEx (*Battle et al., 2017*) consortia. The derived models offer computational predictions of the likely regulatory effects of genomic variation based on disruption or creation of regulatory motifs discovered by the CNNs. While these multi-tissue methods offer an attractive, generally applicable, framework for variant prioritization, they may be missing nuances of the tissue-specific regulatory grammar, and may not be optimal for predictions of regulatory effects that are specific to disease-relevant tissues.

In the present study, we trained CNNs on a broad collection of genome-wide epigenomic profiles capturing chromatin regulatory features from human pancreatic islets, and applied the resulting models to predict the regulatory effects of sequence variants associated with T2D. We demonstrate that these tissue-specific CNN models recapitulate regulatory grammar specific to pancreatic islets, as opposed to discovering regulatory motifs common across multiple tissues. We apply the CNN models to predict islet regulatory variants among the credible sets of T2D-association signals, and demonstrate how CNN predictions can be integrated with genetic and functional fine-mapping approaches to provide single-base resolution of functional impact at T2D-associated loci.

## Results

### CNNs achieve high performance in predicting islet chromatin regulatory features

We collected 30 genome-wide epigenomic profiling annotations from human pancreatic islets, and their FACS-sorted cell subsets from previously published studies (*Bhandare et al., 2010*; *Gaulton et al., 2010*; *Stitzel et al., 2010*; *Parker et al., 2013*; *Pasquali et al., 2014*; *Maher, 2012*; *Thurner et al., 2018*; *Bramswig et al., 2013*; *Ackermann et al., 2016*)(*Supplementary file 1*-STable 1), and re-processed them uniformly with the same computational pipelines. The 1000 bp long genomic sequences encompassing the signal peaks were used, together with vectors representing presence/absence of the 30 islet epigenomic features within these regions, as inputs to train the multi-class prediction CNNs. The resulting CNN models predict presence of these 30 features within any 1000 bp long genomic sequence (*Figure 1—figure supplement 1*). Since the weights in neural network training are initialized randomly and then optimized during training, there is a considerable amount of heterogeneity in the predictive scores resulting from different iterations of the same training process, as networks may converge at different local optimal solutions. To improve robustness of results achieved with these models, we trained a total of 1000 CNNs with 10 different sets of hyperparameters differing in numbers of convolutional filters and their sizes to account for this stochastic heterogeneity (*Supplementary file 1*-STable 2).

Overall, CNNs achieved high performance in predicting the islet epigenomic features in sequences withheld from training, though we observed that the performance varied depending on the predicted feature (*Figure 1*, *Figure 1—figure supplement 2*). The best predictive performance was achieved for features related to promoters, transcription factor (TF) binding and DNA accessibility,

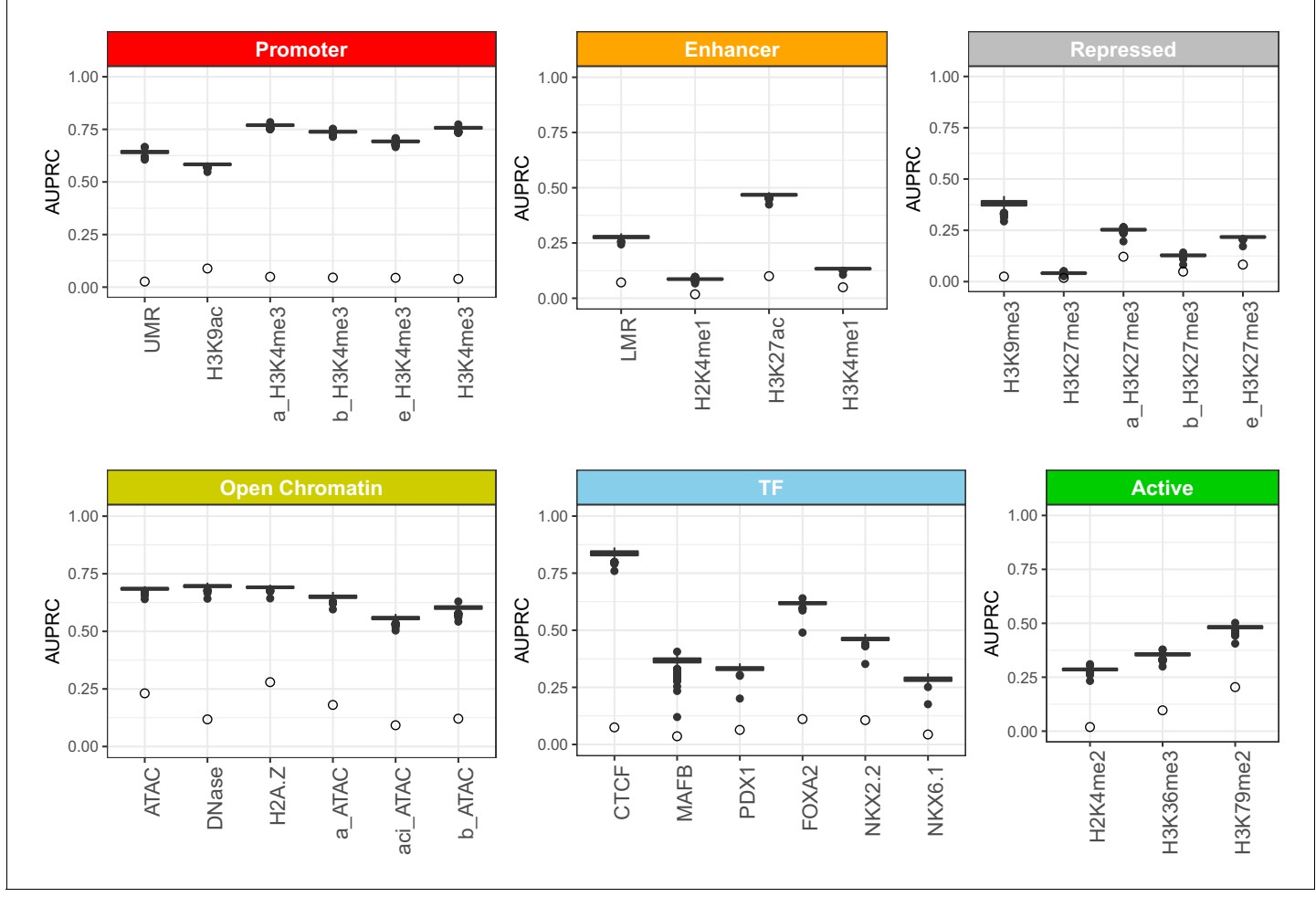

**Figure 1.** Area under precision-recall curves (AUPRC) for 30 islet epigenomic features predicted by CNN models. The AUPRC values were calculated based on performance on the test set formed by 1000 bp sequences from chr2, held out from training and validation. The boxplots show summary of performance across 1000 individual CNN models, and are grouped by corresponding regulatory element. As the interpretation of AUPRC values depend on how well balanced the dataset it, we denote the class imbalance (equivalent to prediction of a random model) for each feature as open circles, which corresponds to the proportion of sequences with the given feature present. Features marked as 'a_", 'b_","e_", and 'aci_", were assayed in FACS-sorted cell populations rather than whole pancreatic islets, and correspond to alpha cells, beta cells, exocrine cells, and acinar cells, respectively (*Bramswig et al., 2013*; *Ackermann et al., 2016*).

The online version of this article includes the following figure supplement(s) for figure 1:

**Figure supplement 1.** Schematic representation of the applied convolutional neural network architecture, with sizes and numbers of filters, and width of pooling indicated for a representative combination of tested hyperparameters.

**Figure supplement 2.** Area under receiver-operator curves (AUROC) for 30 islet epigenomic features predicted by CNN models.

**Figure supplement 3.** Influence of the size of filters in the first convolutional layers on filters' annotation and filter's influence on predictions.

with mean areas under receiver-operator curves (AUROC) of 0.95, 0.89 and 0.89, respectively. Histone mark features associated with active, coding regions, repressed regions and enhancers proved more difficult to predict based on their underlying genomic sequence, with mean AUROCs of 0.82, 0.79 and 0.77, respectively.

In addition to area under receiver-operator curve (AUROC)(*Figure 1—figure supplement 2*), which was used to test predictive performance during CNN training, we also inspected the area under precision-recall curves (AUPRC)(*Figure 1*), a more appropriate measure when predicted classes are not well balanced. We observed that, while variable between features, the predictive performance of the CNN models for the chromatin regulatory features was high, and for the majority of predicted features far exceeded the performance of random predictors. Again, we observed most accurate predictions for promoters, open chromatin and TF binding features, with AUPRCs of 0.70,

0.65 and 0.48, respectively, while enhancers, as well as active, and repressed regions had lower accuracy, with mean AUPRCs of 0.24, 0.37 and 0.20, respectively.

## Convolutional filters recover binding motifs of TFs with roles in pancreatic development

Convolutional filters of the first network layer capture local sequence patterns and motifs, aiding in predictions of islet regulatory features. We hypothesized that many of these would correspond to binding motifs of transcription factors (TFs) with roles in pancreatic islet development and function. For each convolutional filter of the first CNN layer, we derived a position weighted matrix (PWM) based on the observed nucleotide frequencies activating the filter, and compared them to a database of known TF binding motifs. We observed that the number of annotated motifs per network was positively correlated with the size of convolutional filters within the first layer, while the number of filters informative for the predictions (activation standard deviation >0) decreased with filter size (*Figure 1—figure supplement 3*). On average, only 29 out of 320 filters in first CNN layer were annotated to known TF binding motifs, but an average of 177 filters were informative for predictions, indicating that CNN models identify potentially novel sequence motifs not currently represented in the databases of known TF binding motifs.

In total, we identified 373 recurrent annotated binding motifs with <5% false-discovery rate (FDR) sequence similarity to the filters of the first CNN layer, which were detected in >50 networks (*Supplementary file 1*-STable 3). The 10 most frequently discovered non-redundant motifs are listed in *Table 1*. As expected, among the consistently-detected transcription factor motifs, we found motifs for all the transcription factors included in the ChIP-seq training datasets (CTCF, FOXA2, PDX1, MAFB, NKX2.2, NKX6.1). Additionally, CNNs discovered, *ab initio*, binding motifs for several TFs that are known to be important for pancreatic development and for the maintenance of beta and alpha cell functions, including RFX6, HNF1A and NEUROD1 (*Jennings et al., 2015*; *van der Meulen and Huising, 2015*). This demonstrates that the CNN models of the pancreatic islet epigenome are capable of discovering well-established islet TF motifs *ab initio* from genomic sequences.

## Islet CNN models prioritize T2D-associated variants with regulatory roles in pancreatic islets

The regulatory effects of genomic variants can be approximated by comparing the CNN predictions for genomic sequences including different alleles of the same variant. Here, we applied the islet CNN models for prioritization of T2D-associated variants from a recently published GWAS study (*Mahajan et al., 2018*). This study of ~900,000 cases and controls of European ancestry, identified 403 T2D-risk signals, and performed genetic fine-mapping for 380 of them. We ran CNN predictions for all 109,779 variants included within the 99% credible variant sets for these signals, averaging the regulatory predictions for each variant and feature across 1000 individual CNN models to increase robustness. Variants most likely to influence the islet epigenome were then identified through the cumulative distribution function for the normal distribution, separately for each predicted feature, and the lowest q-value for any of the features was assigned to a variant to signify its overall regulatory potential.

We identified a total of 11,389 variants with a q-value <0.05 for any of the 30 features, approximately 10% of the total number of credible set variants. Broadly similar numbers of regulatory variants fell within the different groups of chromatin features (*Figure 2A*). Variants with q < 0.05 were significantly more likely to be evolutionarily conserved than the credible set variants with overall q > 0.05, as assessed by one-sided Wilcoxon rank sum test of the GERP scores (*Cooper et al., 2005*) (p=7.3e-04).

To further validate the functional inference from the islet CNN, we compared the CNN predictions at the credible set variants with results from a recent cis-eQTL study of human islet samples from 420 donors performed by the InsPIRE consortium (*Viñuela et al., 2019*). In this analysis, 91 of the 403 T2D GWAS signals from the largest published meta-analysis (*Mahajan et al., 2018*) overlapped an islet eQTL (defined as p<10e-08): of these, we identified overlapping regulatory variants (as predicted by the CNN q < 0.05) at 73 (~80%). We considered the enrichments of variants predicted to affect the six different groups of chromatin features, and found that variants predicted to

**Table 1.** 10 non-redundant transcription factors binding motifs most frequently detected by first layer convolutional filters at FDR < 5%.

Sequence logos of representative CNN filters are shown. Transcription factor binding motifs redundancy was removed with Tomtom motif similarity search with other motifs detected by CNNs with q < 0.05 for similarity to the main motif are listed in the last column; only three motifs with highest similarity are listed.

| Motif name/TF | Representative CNN filter logo | Motif logo | CNNs with filter match q < 0.05 | Similar TF motifs discovered |
|---|---|---|---|---|
| M6114_1.02 FOXA1 | | | 838 | M6234_1.02 FOXA3, M6241_1.02 FOXJ2, M4567_1.02 FOXA2... |
| M4427_1.02 CTCF | | | 833 | M4612_1.02 CTCFL |
| M1906_1.02 SP1 | | | 677 | M2314_1.02 SP2, M6482_1.02 SP3, M6535_1.02 WT1... |
| M2296_1.02 MAFK | | | 629 | M4629_1.02 NFE2, M4572_1.02 MAFF, M4681_1.02 BACH2... |
| M2292_1.02 JUND | | | 571 | M4623_1.02 JUNB, M2278_1.02 FOS, M4619_1.02 FOSL1... |
| M1528_1.02 RFX6 | | | 556 | M4476_1.02 RFX5, M1529_1.02 RFX7, M5777_1.02 RFX4... |
| M4640_1.02 ZBTB7A | | | 530 | M6539_1.02 ZBTB7B, M6552_1.02 ZNF148, M6422_1.02 PLAGL1... |
| M1970_1.02 NFIC | | | 484 | M5664_1.02 NFIX, M5660_1.02 NFIA, M5662_1.02 NFIB |
| M2277_1.02 FLI1 | | | 442 | M6222_1.02 ETV4, M2275_1.02 ELF1, M5398_1.02 ERF... |
| M6281_1.02 HNF1A | | | 418 | M6282_1.02 HNF1B, M6546_1.02 ZFHX3 |

affect promoter or enhancer activities in islets were preferentially enriched among the top pancreatic islet eQTL results (*Figure 2B*).

We hypothesized that CNN-predicted regulatory variants would also be more likely to show allelic imbalance in chromatin accessibility. We used ATAC-seq data from a previously-published dataset of 17 human pancreatic islets (*Thurner et al., 2018*) to identify 137 pancreatic islet chromatin

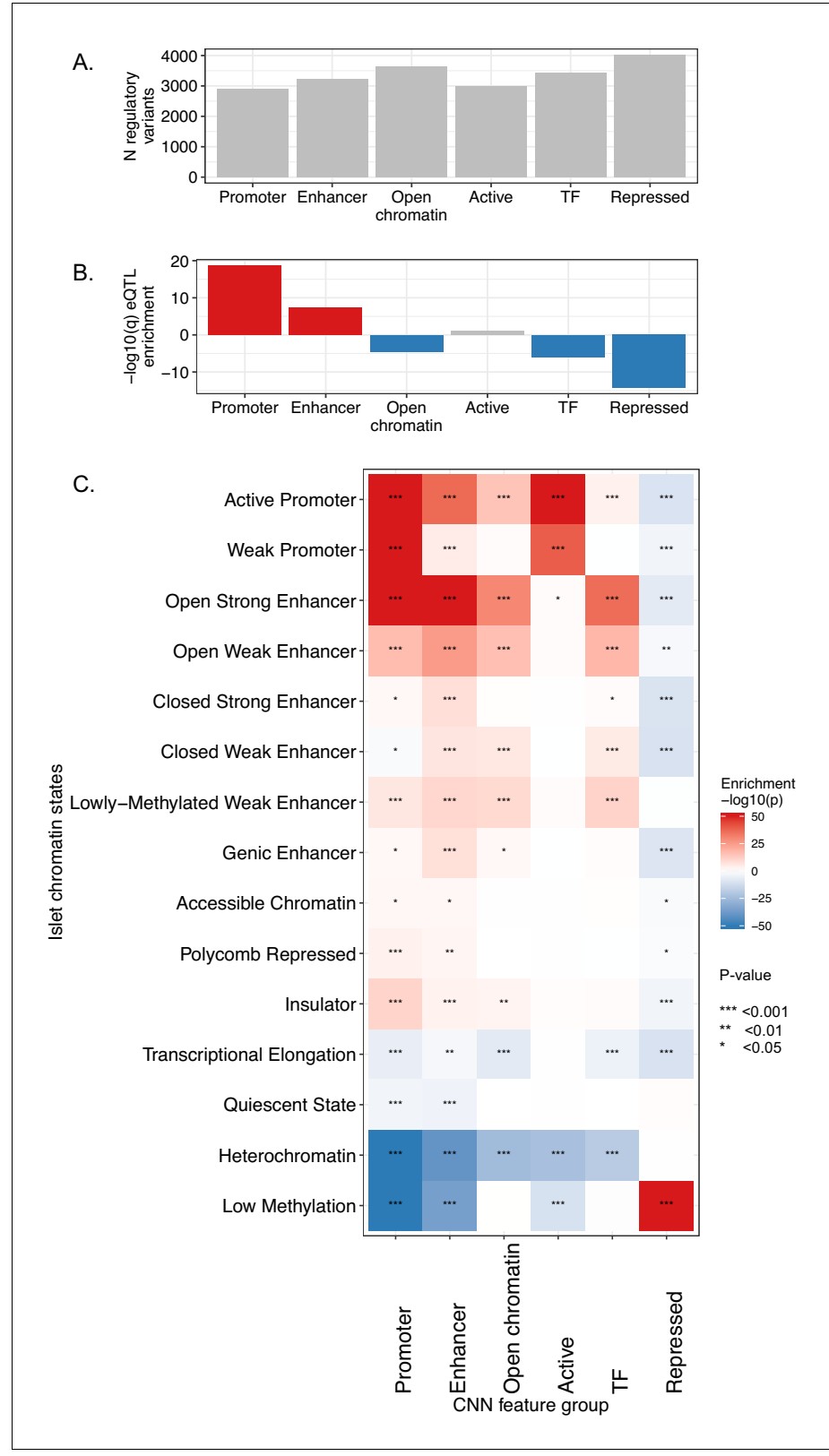

**Figure 2.** Functional characterization of CNN-predicted regulatory variants. (**A**) Distribution of CNN-predicted regulatory variants (q < 0.05) in the six broader CNN feature groups. (**B**) Enrichment of variants predicted to affect the CNN feature groups within variant list ranked by eQTL result p-value from the InsPIRE study (**Viñuela et al., 2019**). Enrichment was calculated with R package gage (**Luo et al., 2009**), red bars indicate gene-set enrichment

*Figure 2 continued on next page*

*Figure 2 continued*

at the top (and blue at the bottom) of the eQTL p-value -ranked list of variants. (**C**) Predicted regulatory variants reside in regulatory elements they are predicted to affect. For each variant we found the lowest CNN q-value among feature groups corresponding to different regulatory elements (promoters, enhancers, open chromatin, active regions, TF binding, repressed regions) predicted from genomic sequence, and we ranked all variants according to these six q-values. We then tested whether variants residing in each of the 15 pancreatic islet chromatin states (*Thurner et al., 2018*) were enriched at the top or bottom of these ranked lists using gene-set enrichment analysis implemented in the R package gage (*Luo et al., 2009*). Colours in the heatmap represent the strength of the enrichment expressed as log10-transformed enrichment q-values, with red colours representing enrichments at the top (enrichment), and blue at the bottom of the ranked lists (depletion). For plotting purposes all -log10(p-values) below −50, or above 50 were truncated to these values. Stars denote significant enrichments: *<0.05, **<0.01 and ***<0.001. Variant level functional annotations and CNN predictions for the credible set variants are available as *Figure 2—source data 1*.

The online version of this article includes the following source data for figure 2:

**Source data 1.** Summary of CNN predictions for all variants from T2D GWAS credible sets.

accessibility QTLs (caQTLs) among the credible set variants, and found these to be significantly enriched among the variants with the lowest CNN q-scores (p=6.9e-20).

Finally, we reasoned that variants predicted to affect function of specific regulatory elements would be more likely to reside within them (e.g. a variant predicted to disrupt enhancer function should be residing in an enhancer region). We tested this by investigating overlap of predicted regulatory variants with human pancreatic islet chromatin state maps (*Thurner et al., 2018*). For this purpose, we considered the lowest q-values within each of the earlier described groups of regulatory features, corresponding to promoters, enhancers, open chromatin, transcription factor binding, as well as active and repressed regulatory regions (*Figure 2C*). Overall, we observed good agreement between the predicted disrupted regulatory elements and the variant located within them. Additionally, we observed a depletion of regulatory variants within heterochromatin and other low methylation sites.

These findings provide validation that variants predicted, on the basis of the CNN, to have the greatest impact on islet regulatory function, were enriched for overlap with regions previously characterized of particular importance for the regulation of islet transcription. However, the regulatory signals captured by cis-eQTL analyses have the disadvantage that they often provide poor localization of the regulatory variant to patterns of local LD. Similarly, chromatin state data lack resolution beyond the 200 bp intervals used for data aggregation. In contrast, the predictions from CNN models offer single-nucleotide resolution.

## Convergence between CNN predictions and fine-mapping approaches

If the CNN models are correctly identifying regulatory variants, we would expect to see convergence between variants predicted to have regulatory effects based on the CNNs, and those assigned high genetic posterior probabilities of association (gPPA) from genetic fine-mapping. To test for such convergence, we generated the null distribution of randomly distributed regulatory variants through 1000 permutations of CNN q-values, while preserving the structure of credible sets at the 380 T2D-associated signals. We observed enrichment of islet regulatory variants (CNN q < 0.05) among variants with highest PPAs (*Figure 3A*), compared to the permutation-based random distribution of regulatory variants (p=0.001). Overall, we found that 28.8% of variants with gPPAs > = 0.8 had predicted regulatory effects with q < 0.05.

It is standard to complement genetic fine-mapping with information from the genome-wide enrichment of association signals within regulatory annotations in disease-relevant tissues, deriving functional posterior probabilities of association (fPPAs) that combine genetic and epigenomic insights into variant function (*Huang et al., 2017*). As with the gPPAs, we observed that variants with high fPPAs, obtained through incorporating enrichments in regulatory elements from human pancreatic islet chromatin state maps (*Thurner et al., 2018*), were enriched for CNN-predicted regulatory effects (p=0.001)(*Figure 3B*). Overall, 40.6% of variants with fPPAs > = 0.8 had predicted regulatory effects with q < 0.05.

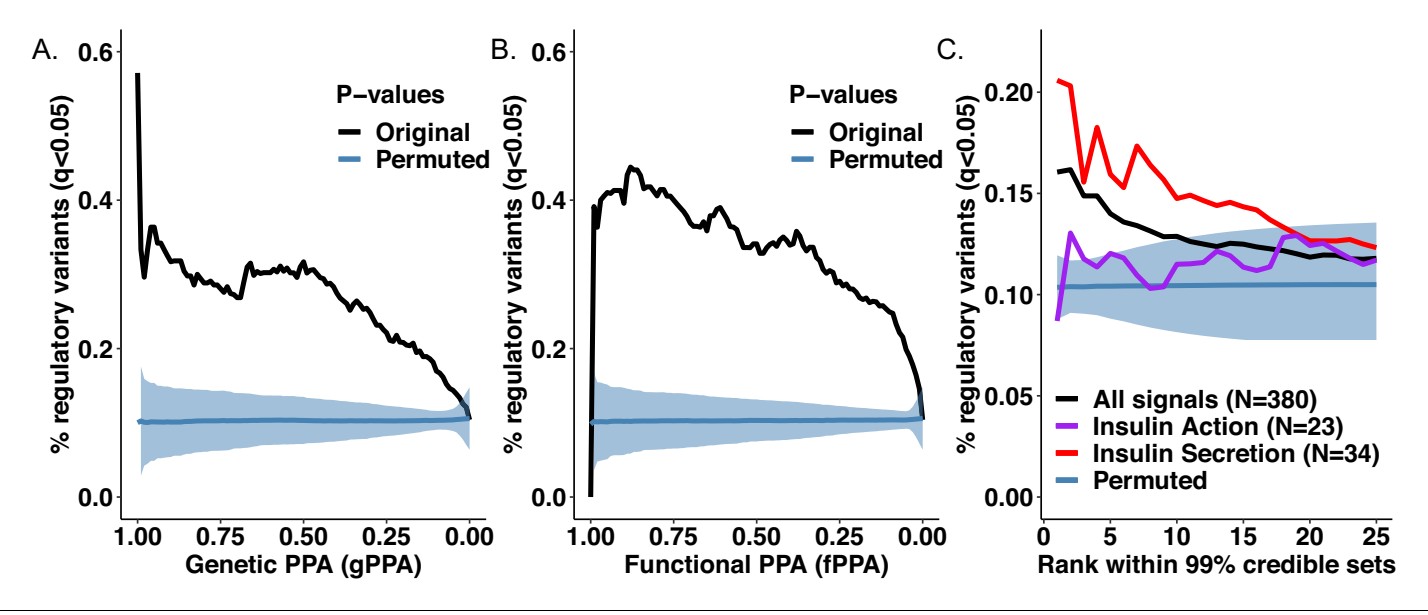

**Figure 3.** Convergence between CNN regulatory predictions and fine-mapping approaches for functional variant prioritization. (**A**) Regulatory variants (black) are enriched among variants with highest genetic PPAs (gPPAs) over permuted background (blue). (**B**) Regulatory variants (black) are enriched among variants with highest functional PPAs (fPPAs) generated with FGWAS over permuted background (blue). (**C**) Regulatory variants (black) are enriched among variants with top PPA ranks within 99% sets of credible variants over permuted background (blue), as well as at top ranks of signals acting through insulin secretion (red) over insulin action (purple) mechanisms.

The online version of this article includes the following figure supplement(s) for figure 3:

**Figure supplement 1.** Comparison of CNN regulatory predictions made with the islet-specific CNN ensemble to predictions made with the publicly available DeepSEA model.

As fine-mapping resolution varies between signals, we conducted analogous analyses based on variant rank within each credible set, irrespective of the quantitative PPA value (*Figure 3C*). Again, we observed higher proportions of predicted regulatory variants at higher PPA ranks within fine-mapped loci, when compared to random distribution of regulatory variants.

One might expect that CNN models trained with pancreatic islet epigenomic annotations would display the strongest evidence for prediction of regulatory variants at the subset of T2D GWAS signals characterized by defects in insulin secretion (*Dimas et al., 2014*; *Wood et al., 2017*), signals that are likely mediated through events in pancreatic islets. Indeed, we observed that the enrichment of islet CNN-regulatory variants was more marked within the top ranks of insulin secretion signals. In contrast, T2D signals characterized by a primary defect in insulin action (which typically involve mechanisms in liver, fat and muscle) showed no enrichment over the permuted background (*Figure 3C*).

Collectively, these data corroborate the convergence of the agnostically-derived CNN variant regulatory scores and diverse measures of islet biology, and indicate the potential for CNNs to support causal variant prioritization in T2D. They also emphasize the value of functional analyses that take account of the tissue-specificity of both transcriptional regulation and disease pathogenesis.

## Islet CNN models correctly identify known functional variants at T2D-associated loci

We tested whether the CNN regulatory predictions for individual variants can be integrated with previous fine-mapping approaches to further resolve T2D-associated signals. Overall, we found that, at 327 out of the 380 fine-mapped signals from the most recent European T2D GWAS study (*Mahajan et al., 2018*), there was at least one predicted regulatory variant (defined as q < 0.05). Among the 74 signals previously fine-mapped to a single variant (with gPPA or fPPA > = 80%), we found 28 variants predicted to be regulatory in pancreatic islets (*Supplementary file 1*-STable 4), in line with the previously reported overall enrichment of T2D-association signals in the regulatory

elements specific to islets (*Parker et al., 2013*; *Pasquali et al., 2014*; *Thurner et al., 2018*; *Miguel-Escalada et al., 2018*). These included two well-studied variants (rs10830963 at *MNTR1B*, rs7903146 at *TCF7L2*), both previously shown to alter enhancer activities (*Gaulton et al., 2015*; *Gaulton et al., 2010*): these served as positive controls for the application of CNNs to functional variant prioritization.

## Islet CNN models refine regulatory mechanisms at T2D-associated loci

While functional fine-mapping can be invaluable in narrowing down the list of most likely causal variants through investigating overlaps with regulatory elements in appropriate tissues, this strategy may not provide sufficient resolution at loci where several variants reside in the same regulatory element. Among the credible sets of variants, we identified 93 signals featuring at least two variants with fPPAs > = 20%, indicating that, even after functional fine-mapping, there is no unambiguously causal single variant. At 37 of these 93 signals, CNNs predicted islet regulatory variants among the top fPPA candidates (*Supplementary file 1*-STable 5). At 25 signals, integration of CNN regulatory predictions downstream of the functional fine-mapping highlighted a single most likely causal variant, with either just a single islet regulatory variant predicted among the credible variants, or the top regulatory variant having much lower q-value (difference in -log10(q) > 100) than other predicted regulatory variants at the signal (a few such signals are highlighted in *Figure 4* and *Figure 5*), narrowing down the list of candidates for further functional follow-up studies.

To explore this further, we focused on a T2D-association signal at the *PROX1* locus, identified after conditioning the T2D association on the primary (most significant) association signal at rs340874; on the basis of patterns of phenotypic association of the T2D-risk allele with continuous diabetes-related traits, this variant is associated with a primary effect on insulin secretion (*Dimas et al., 2014*). This conditional signal has been fine-mapped to two plausible variants, rs79687284 and rs17712208, on the basis of genetic and genomic data (*Figure 5A*). These variants are in perfect LD ($R^2$ = 1.0, D'=1.0) and are located 376 bp apart within the same open strong enhancer in islets. Neither genetic fine-mapping (in European populations) nor functional fine-mapping were able to further resolve the association at this signal. When we investigated pancreatic islet ATAC-seq data from four individuals heterozygous for these two variants (as confirmed by array genotype data), we observed strong allelic imbalance at both variants (rs17712208 p=1.55e-06; rs79687284 p=6.10e-05) (*Figure 5B*). This supports regulatory effects of this *PROX1* signal in the pancreatic islets, but highlights the difficulties in resolving the causal variant. These data are also consistent with the possibility that both variants are contributing to the functional effect.

At this signal, both variants were scored as potentially regulatory by CNNs with q < 0.05, but the q-value at rs17712208 was much lower (q = 1.69e-160) highlighting this variant as the more likely regulatory candidate in the islets. This q-value corresponded to a prediction that the T2D risk A-allele would lead to a significant reduction in the H3K27ac mark ($\triangle$H3K27ac = −0.12) (*Figure 5C*) indicative of an active regulatory element presence. The second variant, rs79687284, was assigned a much less remarkable q-value (q = 0.002) and only predicted to affect promoter features, with the top predictions pointing to a mild reduction of H3K4me3 mark in alpha and beta cells ($\triangle$a_H3K4me3 = −0.01, $\triangle$b_H3K4me3 = −0.01).

While it remains plausible that rs79687284 plays a regulatory function in a specific cellular context, the functional annotation and CNN predictions point to rs17712208 as the more likely causal variant (particularly given that both map to an enhancer rather than a promoter). We performed an in silico saturated mutagenesis for the rs17712208 variant (*Figure 5D*) to identify the regulatory sequence motifs at this locus affected by the variant. We observed that the reference T-allele is a crucial nucleotide in the HNF1B binding motif (*Figure 5E*), and that introduction of an A-allele at this position disrupts this motif, leading to predicted loss of the H3K27ac mark. We note that both *PROX1* and *HNF1B* are co-expressed during pancreatic development in a previously published iPSC-derived model of beta cell differentiation (*Perez-Alcantara et al., 2018*).

To obtain experimental confirmation, we assessed the transcriptional activity of both variants in the human EndoC-βh1 beta-cell line (*Ravassard et al., 2011*). This is a widely-used model of beta-cell function, which expresses both *PROX1* and *HNF1B,* and represents the majority cell type in the islets. Using an in vitro reporter assay, we confirmed that the A-allele at rs17712208 resulted in significant repression of enhancer activity (p=0.0001), while no significant change was observed for the T2D-risk allele at rs79687284 (p=0.15)(*Figure 5F*).

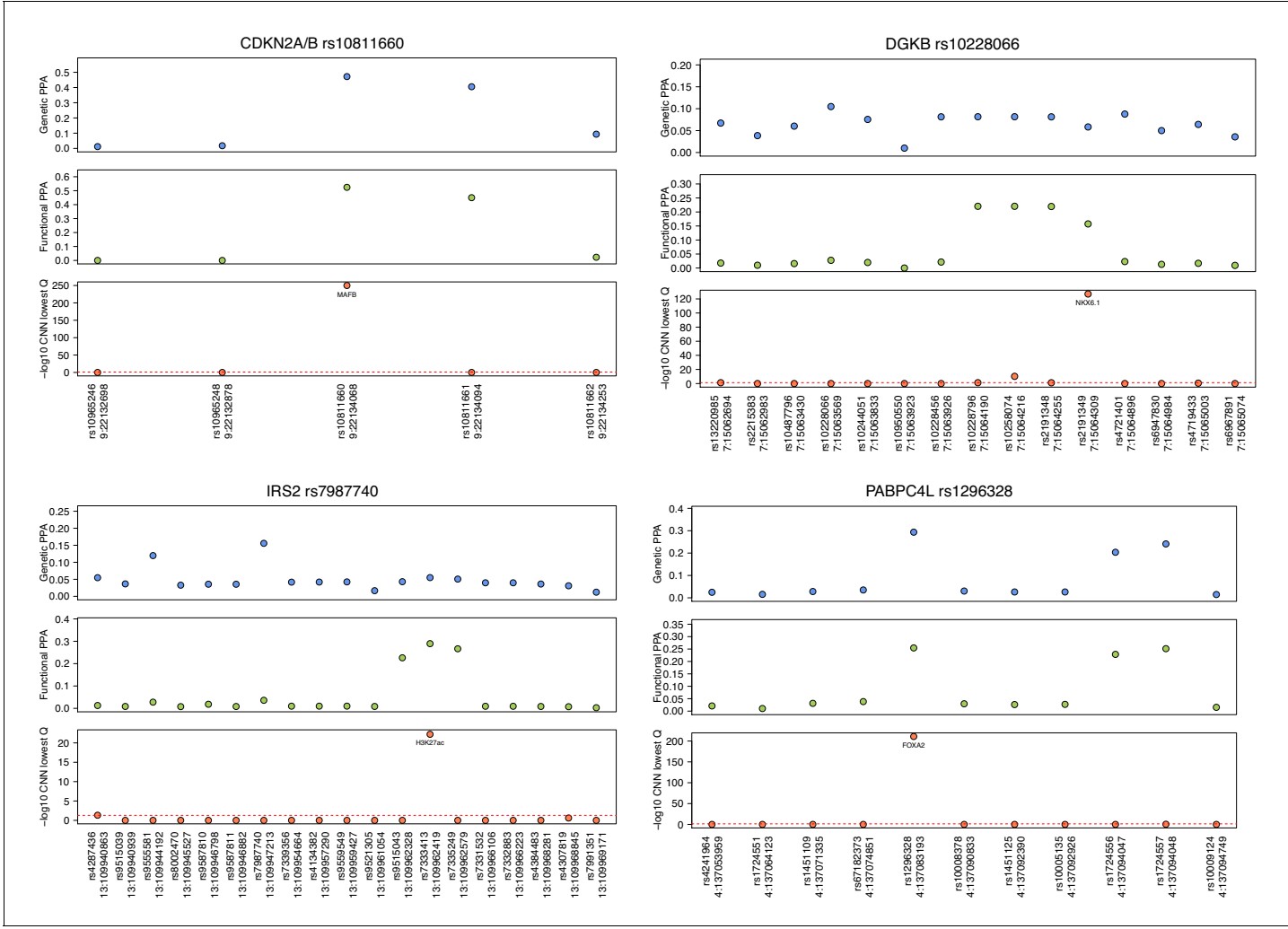

**Figure 4.** Examples of T2D-association signals where integration of CNN regulatory variant prediction downstream of functional fine-mapping refines the association signals to single candidate variants. Genetic PPAs (gPPAs) are shown in the top panels as blue points, functional PPAs (fPPAs) are shown in the middle panels as green points, and -log10-transformed q-values from CNN predictions are shown in the bottom panels as red points.

These examples illustrate how diverse approaches to variant prioritization at the T2D-associated loci show strong convergence towards the same candidate variants, and how CNN models can complement fine-mapping approaches in providing single-base resolution to refine the association signals. Coupled with additional evidence that these signals exert their mechanism of action in the pancreatic islets, rather than other tissues implicated in T2D aetiology, the predicted regulatory variants may provide attractive targets for further functional follow-up studies.

## Comparison of tissue-specific with multi-tissue CNN model

Finally, we compared the regulatory variant predictions made with the pancreatic islet tissue-specific CNN models described in this study with predictions made with a widely-used multi-tissue variant prioritization tool, DeepSEA (*Zhou and Troyanskaya, 2015*) (*Figure 3—figure supplement 1 - C*). We compared our results to the multi-tissue significance scores reported by DeepSEA for each variant, as well as to the predicted effects on pancreatic islet chromatin accessibility (based on ENCODE DNase-seq data ['PanIslets'] generated from primary pancreatic islet cells, one of the tissues contributing to the overall multi-tissue score). Overall, we observed a modest but highly significant correlation of predictions with both these regulatory prediction scores reported by DeepSEA (multi-tissue: r = 0.227, p<2.2e-16; PanIslets: r = 0.223, p<2.2e-16) (*Figure 3—figure supplement 1 - AB*).

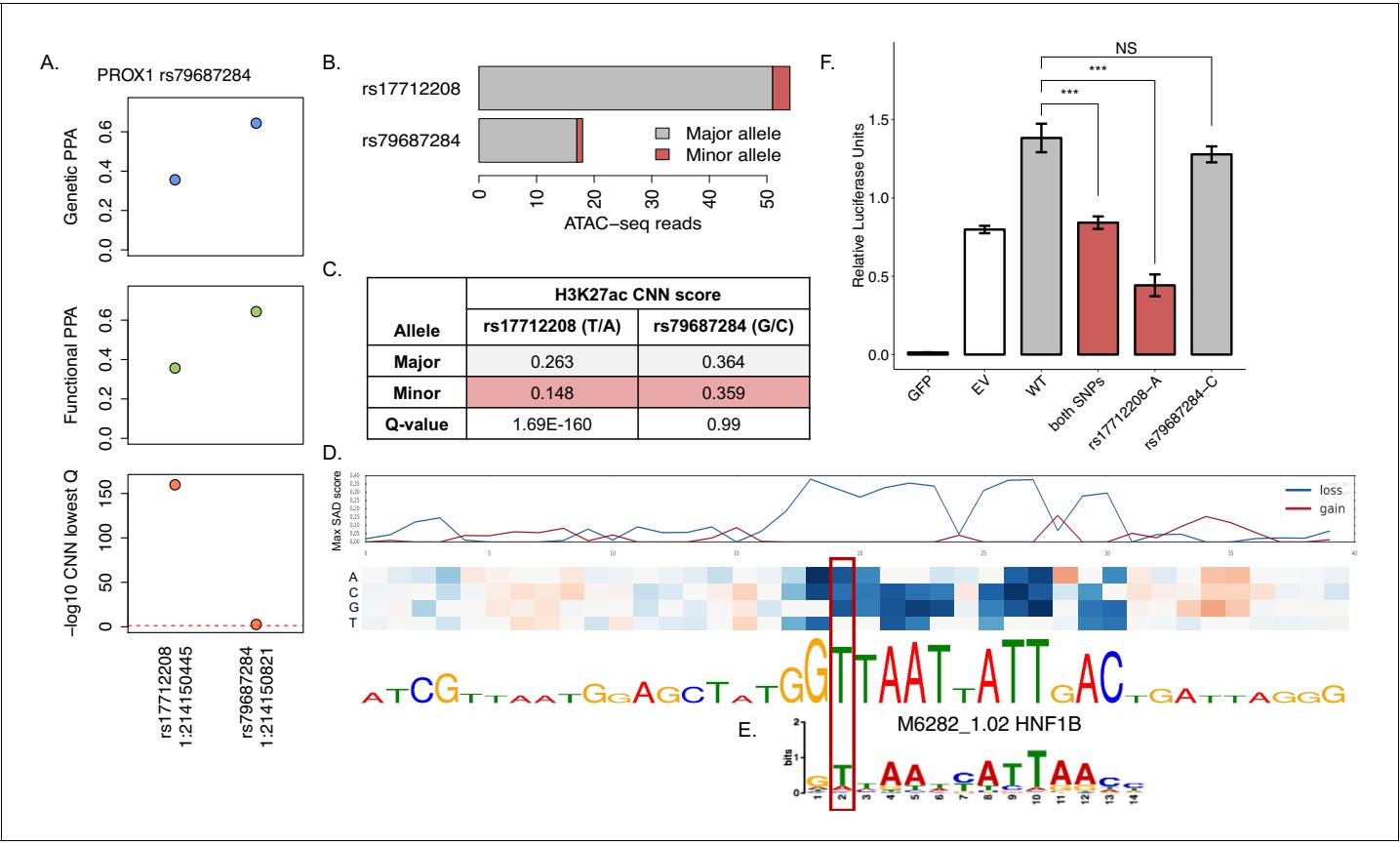

**Figure 5.** CNN regulatory predictions help refine the association signal at PROX1 locus, previously fine-mapped to only two variants: rs17712208 and rs79687284. (A) Genetic PPA (gPPA), functional PPA (fPPA) and -log10(q-value) of the CNN islet regulatory predictions for both variants. (B) Allelic imbalance in open chromatin across four pancreatic islets heterozygous for the variants. Allele counts for the major (grey) and minor (red) alleles are shown for both variants. (C) Table summary with CNN predictions for the H3K27ac mark for both variants. (D) In silico saturated mutagenesis for 40nt flanking sequence around the rs17712208 variant for the H3K27ac predictions. The line plots in the upper panel indicate the SAD (SNP accessibility difference) scores corresponding to absolute highest values from the heatmap below, with blue line indicating loss of function, and red – gain of function changes. Blue fields in the heatmap indicate that a given nucleotide substitution would result in decrease in prediction values for H3K27ac, while red field indicate increase in the predictions. The height of letters in the sequence below the heatmap indicated the relative importance of each nucleotide in the final predictions. (E) Matched HNF1B binding motif is shown below. (F) Luciferase reporter assays confirmed that the A allele of rs17712208 resulted in significant repression of enhancer activity, while no effect was observed for the rs79687284 variant. GFP = green fluorescent protein (negative control), EV = empty vector (baseline). Source file with luciferase intensity values is available as *Figure 5—source data 1*.

The online version of this article includes the following source data for figure 5:

**Source data 1.** Luciferase intensity values.

As with our islet CNN models, we observed strong enrichment of predicted regulatory variants amongst the top ranks of the T2D GWAS 99% credible variant sets for both modes of DeepSEA, as compared to permuted background (*Figure 3—figure supplement 1 - C*). However, both the multi-tissue and PanIslets DeepSEA models performed better within the subset of T2D signals acting through insulin action rather than through insulin secretion signals, predicting higher proportions of regulatory variants at top ranks of credible sets for the former, contrary to our expectations for the PanIslets model (*Figure 3—figure supplement 1 - D*). In addition, the DeepSEA framework failed to predict regulatory impact at loci where T2D causative variants have been established through multiple lines of evidence (such as at rs10830962 at *MTNR1B* (*Gaulton et al., 2015*) and rs7903146 at *TCF7L2 Gaulton et al., 2010*), both of which were correctly identified as regulatory in our islet-based study. Similarly, while the rs17712208 variant at *PROX1* reported in this study was found to be regulatory using the multi-tissue mode of DeepSEA (p=6.25e-03), the predictions for pancreatic islets accessibility profiles (PanIslets) were only borderline significant for this variant (p=0.049).

Our in silico saturated mutagenesis analysis indicated NEUROD1 as the likely motif created at the *MTNR1B* locus (not shown), and HNF1B as the motif disrupted at *PROX1* (*Figure 5E*). These transcription factors are critical for the function of pancreatic islet beta cells (*Gu et al., 2010*), and likely represent tissue-specific processes, more liable to be missed in a multi-tissue model. We anticipate that CNN models trained on data from multiple tissues may be biased towards sequence features and motifs present across all the tissues, and may not detect tissue-specific signals with high confidence. Given the demonstrated enrichment of T2D association signals in regulatory elements specific to the pancreatic islets, we demonstrate the value of training the prediction models in the tissues most relevant to the phenotype in question, at least for those loci where the tissue-of-action can be defined.

## Discussion

Application of deep learning methods to characterize the regulatory potential of noncoding variants has been a subject of interest in recent years. Instead of predictions across an array of tissues, in the current study we focused on multiple genome-wide epigenomic annotations available for pancreatic islets, one of the central tissues impacting T2D aetiology. We demonstrate how the CNN regulatory predictions for genomic variants can be integrated downstream of fine-mapping approaches for further refinement of GWAS association signals.

Overall, we observed that CNNs were capable of learning the regulatory code from the underlying genomic sequence, though the prediction accuracy differed among the studied epigenomic features. The individual feature predictive performance was likely affected by a number of factors, in particular the quality of input data, and the total number of peaks called for a feature, as deep learning methods require many training examples to learn to generalize well. Furthermore, epigenomic features differ in their characteristics, including peak width and specificity of sequence motifs predictive of their presence. Thus, when training a multi-task prediction network, we may expect that network parameter optimization might not work equally well for disparate features. Promoter regions, DNA accessibility and binding of specific transcription factors proved the easiest to predict, as these features are characterized by very distinct sequence motifs that can be identified by the network's convolutional filters. Importantly, the binding motifs learned de novo from the genomics sequences of signal peaks presented to the networks included several transcription factors with well-established functions in pancreatic development and function.

In the present study, we trained 1000 individual CNN models with different hyperparameters to aid in prioritization of the likely functional variants from the largest T2D GWAS study to date (*Mahajan et al., 2018*). We observed an overall convergence of the genetic fine-mapping, functional fine-mapping and regulatory predictions generated by the tissue-specific CNN models, as evidenced by enrichment of predicted functional variants among variants with highest gPPAs, fPPAs and PPA ranks. To our knowledge, this is the first time such enrichment has been reported for in silico predictions of regulatory variants. Our findings contrast with those from a recent study, which argued that computational methods are not yet mature enough to be applied in GWAS fine-mapping (*Liu et al., 2019*).

In our study, we have observed multiple examples of convergence of the top scoring variants by CNNs with the top variants identified through functional fine-mapping using islet chromatin state maps. We also note that the CNN-prioritized variants were significantly enriched in islet chromatin accessibility QTLs, as well as variants residing in islet functional regulatory elements, including promoters and strong open enhancers, which is in line with findings of previous functional genomics studies (*Thurner et al., 2018*; *Parker et al., 2013*; *Pasquali et al., 2014*). We demonstrate that when used in conjunction with high-resolution genetic or functional fine-mapping, CNN predictions can provide a powerful way of dissecting the causal variant from a set of T2D-associated variants. The results presented in this study will facilitate further functional follow-up studies to fully elucidate the mechanisms underlying the associated disease susceptibility at the prioritized variants.

One important limitation of this approach is that the value of applying tissue-specific CNN models to guide the identification of causal variants depends on knowing the effector tissue for the signals in question. With complex diseases, such as T2D, this is not always straightforward, as multiple tissues (including pancreatic islets, liver, adipose and skeletal muscle) have been implicated in T2D aetiology. Additionally, even within the correct tissue, inference can be complicated by the fact that

tissues represent heterogeneous mixtures of cell populations, of which only a subset might be directly relevant to disease aetiology. The growing availability of cell type specific functional genomic datasets will make it possible to explore the extent to which this more precise assignment of disease pathology allows more accurate inference regarding the causal mechanisms.

The development of high-throughput experimental methods for assaying regulatory variants, such as massively-parallel reporter assays (MPRAs), offers a complementary route to deriving information on variant-specific function. However, these methods are subject to their own sets of biases and errors, and genome-wide deployment can be problematic. More recent methods – such as HiDRA (high-resolution dissection of regulatory activity)(*Wang et al., 2018*) – allow further scale-up, but are limited to sites heterozygous in the assayed sample. Nevertheless, large scale experimental studies such as these provide a further dimension of information that can be folded into future rounds of supervised model training. CNN models have the potential to complement high-throughput experimental approaches to elucidate regulatory variants, by learning the regulatory grammar at genomic locations assayed directly in these experiments, and allowing extrapolation to variants (and cell types) not assayed directly. Indeed, we anticipate that in the near future, deep learning based fine-mapping approaches will become part of routine downstream analyses of GWA data.

## Materials and methods

### Collection and processing of islet data

We collected 30 genome-wide epigenome profiling datasets from human pancreatic islets from previously published studies (*Bhandare et al., 2010*; *Gaulton et al., 2010*; *Stitzel et al., 2010*; *Parker et al., 2013*; *Pasquali et al., 2014*; *Maher, 2012*; *Thurner et al., 2018*) (*Supplementary file 1*-STable 1). Where raw sequencing data was available, we uniformly reprocessed the data using the default settings of the AQUAS Transcription Factor and Histone ChIP-Seq processing pipeline (https://github.com/kundajelab/chipseq_pipeline) and ATAC-Seq/DNase-Seq pipeline (https://github.com/kundajelab/atac_dnase_pipelines), mapping against the human reference genome build hg19, and using MACS2 for peak calling. We used the coordinates of the optimal peaks sets produced with the Irreproducible Discovery Rate (IDR) procedure for further analysis. We grouped the 30 datasets into six broader feature categories: promoters, enhancers, open chromatin, TF binding, active, or repressed regions, based on previous reports of regulatory elements associated with presence of the different chromatin modifications (*Zhou et al., 2011*; *Kundaje et al., 2015*). This grouping was only used for convenience of presentation of the results, and to facilitate their interpretation.

### CNN input sequence and feature encoding

The training and test datasets for the convolutional neural network training were created analogously to the approach used in Basset (*Kelley et al., 2016*). Briefly, 1000 bp genomic intervals were assigned to all called narrow peaks, by extending 500 bp on each side of the centre of the peak. Peaks were greedily merged until no peaks overlapped by >200 bp, and the centre of the merged peak was determined as a weighted average of the midpoints of the merged peaks, weighted by the number of chromatin features present in each individual peak. This resulted in 505,273 genomic intervals of 1000 bp length with assigned presence/absence of the 30 islet epigenomic features, with an average of 2.62 chromatin features per interval. The genomic sequences of the intervals were extracted from the hg19 human reference genome and encoded as one-hot code matrix, mapping the sequences into a 4-row binary matrix corresponding to the four DNA nucleotides at each position. For each 1000 bp sequence, we created an accompanying feature vector denoting which of the 30 datasets showed a signal peak overlapping the sequence. All sequences from chromosomes 1 (N = 43,029, 8.52% total) and 2 (N = 40,506, 8.02% total) were held out from the training and applied as validation and test sets, respectively.

### CNN training

CNN models were trained using code from the open source package Basset (*Kelley et al., 2016*) implemented in the Torch7 framework (http://torch.ch). We trained a total of 1000 CNN models with 10 different hyperparameter settings (Sup. Table 2), differing in sizes and numbers of

convolutional filters applied. We averaged the results across these 1000 models to achieve robust results, and overcome the CNN training stochasticity. Each network contained three convolutional layers, each followed by a rectified linear unit (ReLU) and a max pooling layer, and two fully connected layers. The final output layer produced the predictions for the 30 features. The network architecture is schematically shown in SFig.1. The network training was stopped when the area under receiver-operator curve (AUROC) did not improve in 10 subsequent training epochs. The predictive performance of the networks was assessed by evaluating predictions made on the sequences from chr2 constituting the test set. We calculated the AUROCs and AUPRCs using R package PRROC v.1.3 (*Grau et al., 2015*).

## Sequence motifs captured by convolutional filters

The convolutional filters in the first CNN layer detect repeatedly occurring local sequence patterns, which increase the prediction accuracy. These patterns can be summarized as position-weighted matrices (PWMs), derived by counting nucleotide occurrences that activate the filter to more than half of its maximum value, as implemented in the Basset framework. The resulting PWMs were matched to the CIS-BP database (*Weirauch et al., 2014*) of known transcription factor binding motifs using the Tomtom v.4.10.1 motif comparison tool (*Gupta et al., 2007*). We repeated this for all the 1000 trained networks, and found recurrent motifs by identifying binding motifs matched at FDR < 5% in at least 50 networks. The redundant motifs were filtered out from the top motifs table by comparing all the database motifs against each other using the Tomtom v4.10.1 motif comparison tool, and reporting the top scoring motif as the main hit, and other motifs matching it at FDR < 5% as secondary motifs.

## Application of CNN models to variant prioritization

We obtained the credible variant sets from a recent T2D GWAS study (*Mahajan et al., 2018*). For each variant, we run predictions for the 30 islet epigenomic features with all 1000 trained CNN models for two 1000 bp long sequences: first including the reference allele and second including the alternative allele, with the variant positioned in the middle of the 1000 bp sequence. We calculated the mean differences in prediction scores across all the 1000 CNNs for each of the investigated features between the two sequences for each variant. We then estimated p-values for each of the variants and each feature with the cumulative distribution function of normal distribution, and applied FDR procedure for multiple testing correction. The overall regulatory potential of each variant was quantified as the lowest q-value for any of the 30 features.

## Functional validation of CNN regulatory predictions

To test the validity of the resulting regulatory predictions for the credible set variants, we tested whether there was an enrichment of variants predicted to affect any of the six feature groups among pancreatic islet expression QTLs (eQTLs) from a recent study in 420 individuals (*Viñuela et al., 2019*). We downloaded the full set of results with nominal p-values for gene-level eQTLs, and focused on the results for the credible set variants, for which we also have generated CNN predictions in this study. We applied gene-set enrichment analysis implemented in R package 'gage' v.2.32.1 to calculate enrichments of CNN-predicted regulatory variants within variant list ranked by eQTL p-values.

In a similar manner we tested whether variants with lower q-values are more likely to act as chromatin accessibility QTLs (caQTLs) exhibiting allelic imbalance in open chromatin in the ATAC-seq data from 17 human pancreatic islets (*Thurner et al., 2018*). We identified the islet caQTLs among the credible set variants by investigating variants overlapping the ATAC-seq peaks. We required > = 2 subjects with a heterozygous genotype for the variant, and > = 5 sequencing reads overlapping each allele. We calculated the significance of open chromatin allelic imbalance with the negative binomial distribution test, and used an FDR-adjusted p-value of 0.05 to define the 137 caQTLs. The enrichment of these caQTLs within the credible set variant list ranked by their lowest q-value was calculated using R package 'gage' v.2.32.1.

Finally, we tested whether variants predicted to affect specific regulatory elements (promoters, enhancers, open chromatin, TF binding, active, or repressed regions) were more likely to reside in these regulatory regions in islets, as defined by the high-resolution chromatin state maps of human

pancreatic islets, partitioning the genome into 15 regulatory states based on patterns of chromatin accessibility and DNA methylation integrated with established ChIP-seq marks (*Thurner et al., 2018*). Using the generally applicable gene set enrichment R package gage (version 2.32.1) (*Luo et al., 2009*) we tested whether variants overlapping each chromatin state were more likely to be found at the top or bottom of the variant list ranked by the lowest q-values within each of the feature groups.

## Evaluating convergence between fine-mapping and CNN predictions

To test the overall convergence between the two complementary methods for variant prioritization: genetic fine-mapping and CNN regulatory predictions, we evaluated whether we observed more regulatory variants at highest genetic PPAs. We examined the proportions of regulatory variants predicted by CNNs using q-value <0.05 as a cut-off, at different thresholds of genomic PPA (gPPA) going down from PPA of 1.00 to 0.00 in steps of 0.01, and observed decreasing proportions of regulatory variants with decreasing gPPA. We compared this to a random distribution of regulatory variants with respect to gPPA by permuting the CNN q-values 1000 times, preserving the overall number of significant variants, as well as the number and overall structure of credible sets. The enrichment p-value was calculated from these permutations by comparing the areas under curves with fraction of variants significant at each gPPA threshold generated in each permutation with the area under the curve calculated from the original results. Areas under curves were calculated using the AUC function of the DescTools R package v. 0.99.28 (*Signorelli, 2019*). We repeated the same for functional fine-mapping PPAs (fPPA), using fPPAs generated through incorporation of pancreatic islet regulatory elements defined by chromatin state maps (*Thurner et al., 2018*). To account for the differing degree of fine-mapping resolution at different GWAS loci, we also evaluated whether we observed higher proportion of regulatory variants with q < 0.05 among the variants with the highest PPA within each locus, regardless of the actual PPA value, and repeated the same procedure investigating genomic PPA ranks within each signal. In the same way we evaluated whether we observed lower q-values at the top ranks of signals known to act through insulin secretion mechanisms (N = 34), rather than insulin action (N = 23) (*Wood et al., 2017*; *Dimas et al., 2014*).

## Identification of T2D association signals further refined by incorporation of CNN regulatory prediction

We identified the T2D association signals where the islet CNN models can help with further refinement by investigating signals comprising at least two variants with fPPA > = 0.2. In *Figures 4* and *5* we highlighted signals where CNN predictions point to a single variant, among these fPPA > = 0.2 variants, with a much higher regulatory score than the remaining variants at these loci.

The in silico saturated mutagenesis was performed using the basset_sat.py script from the Basset framework (*Kelley et al., 2016*). The TF binding motifs overlapping the variant site were identified with FIMO from the MEME suite v. 4.11.2 (*Grant et al., 2011*) using the CIS-BP motifs database (*Weirauch et al., 2014*).

## Plasmid transfection and luciferase reporter assay

We experimentally validated the CNN regulatory predictions for the two variants (rs17712208 and rs79687284) at *PROX1* locus with luciferase reporter assay. Briefly, human EndoC-βH1 cells (*Ravassard et al., 2011*) were grown at 50–60% confluence in 24-well plates and were transfected with 500 ng of empty pGL4.23 [luc2/minP] vector (Promega, Charbonnieres, France) or pGL4-minP-PROX_enhancer vectors (wildtype, rs17712208-A and rs79687284-C) with FuGENE HD (Roche Applied Science, Meylan, France) using a FuGENE:DNA ratio of 6:1. The EndoC-βH1 cells (RRID: CVCL_L909) were a purchased from Univercell BioSolutions (http://www.univercell-biosolutions.com/human-heart-cells-and-stem-cells-production) and tested negative for mycoplasm. Primer sequences used for cloning and site-directed mutagenesis (SDM) are listed in *Supplementary file 1*-STable 6. Restriction enzymes NheI and XhoI were used for all subsequent cloning. Luciferase activities were measured 48 hr after transfection using the Dual-Luciferase Reporter Assay kit (Promega). The firefly luciferase activity was normalized to the Renilla luciferase activity obtained by cotransfection of 10 ng of the pGL4.74[hRluc/TK] Renilla Luciferase vector (Promega).

## Comparison of single- versus multi-tissue CNN predictions

Finally, we compared the predictions resulting from our single-tissue pancreatic islet CNN models with the predictions generated with another publicly available multi-tissue variant prioritization tool DeepSEA (*Zhou and Troyanskaya, 2015*). The functional significance scores for each variant (multi-tissue) and the q-value for chromatin effects in the ENCODE PanIslets primary pancreatic islets cells (single tissue) were generated by submitting VCF files of the credible set variants to the DeepSEA web server (date accessed: 9th May, 2018).

## Code availability

Code used to generate the results of this study is available at https://github.com/agawes/islet_CNN.

## Acknowledgements

MT was a Wellcome doctoral student. ALG is a Wellcome Senior Fellow in Basic Biomedical Science. MMcC is a Wellcome Investigator and an NIHR Senior Investigator. Relevant funding support for this work comes from Wellcome (090532, 106130, 098381, 203141, 212259, 095101, 200837, 099673/Z/12/Z), Medical Research Council (MR/L020149/1), European Union Horizon 2020 Programme (T2D Systems), NIDDK (U01-DK105535), NIH (U01-DK105535; U01-DK085545) and NIHR (NF-SI-0617–10090). The views expressed in this article are those of the author(s) and not necessarily those of the NHS, the NIHR, or the Department of Health.

## Additional information

### Competing interests

Mark I McCarthy: Senior editor, *eLife*. MMcC has served on advisory panels for Pfizer, NovoNordisk and Zoe Global, has received honoraria from Merck, Pfizer, Novo Nordisk and Eli Lilly, and research funding from Abbvie, Astra Zeneca, Boehringer Ingelheim, Eli Lilly, Janssen, Merck, NovoNordisk, Pfizer, Roche, Sanofi Aventis, Servier, and Takeda. As of June 2019, MMcC is an employee of Genentech, and a holder of Roche stock. The other authors declare that no competing interests exist.

### Funding

| Funder | Grant reference number | Author |
| --- | --- | --- |
| Wellcome | 099673/Z/12/Z | Matthias Thurner |
| Wellcome | 090532 | Mark I McCarthy |
| Wellcome | 106130 | Anna L Gloyn<br>Mark I McCarthy |
| Wellcome | 098381 | Mark I McCarthy |
| Wellcome | 203141 | Anna L Gloyn<br>Mark I McCarthy |
| Wellcome | 212259 | Mark I McCarthy |
| Wellcome | 095101 | Anna L Gloyn |
| Wellcome | 200837 | Anna L Gloyn |
| Medical Research Council | MR/L020149/1 | Anna L Gloyn |
| Horizon 2020 Framework Programme | T2D Systems | Anna L Gloyn |
| NIH Clinical Center | U01-DK105535 | Anna L Gloyn<br>Mark I McCarthy |
| NIH Clinical Center | U01-DK085545 | Anna L Gloyn |
| National Institute for Health Research | NF-SI-0617-10090 | Anna L Gloyn |

The funders had no role in study design, data collection and interpretation, or the decision to submit the work for publication. The views expressed in this article are those of the author(s) and not necessarily those of the NHS, the NIHR, or the Department of Health.

## Author contributions
Agata Wesolowska-Andersen, Conceptualization, Data curation, Software, Formal analysis, Investigation, Visualization, Methodology, Project administration; Grace Zhuo Yu, Resources, Formal analysis, Validation, Investigation, Methodology; Vibe Nylander, Resources, Methodology; Fernando Abaitua, Resources, Supervision, Validation, Investigation, Methodology; Matthias Thurner, Data curation, Formal analysis; Jason M Torres, Anubha Mahajan, Data curation, Formal analysis, Methodology; Anna L Gloyn, Conceptualization, Resources, Supervision, Funding acquisition, Investigation, Methodology; Mark I McCarthy, Conceptualization, Resources, Supervision, Funding acquisition, Investigation, Methodology, Project administration

## Author ORCIDs
Agata Wesolowska-Andersen (iD) https://orcid.org/0000-0001-8688-2814
Matthias Thurner (iD) http://orcid.org/0000-0001-7329-9769
Jason M Torres (iD) http://orcid.org/0000-0002-7537-7035
Anna L Gloyn (iD) https://orcid.org/0000-0003-1205-1844
Mark I McCarthy (iD) https://orcid.org/0000-0002-4393-0510

## Decision letter and Author response
Decision letter https://doi.org/10.7554/eLife.51503.sa1
Author response https://doi.org/10.7554/eLife.51503.sa2

# Additional files

## Supplementary files
• Supplementary file 1. Supplementary Tables. (**STable 1**) Summary of publicly available datasets used to train the CNN models of human pancreatic islet epigenomic features. Where indicated, the original raw data was reprocessed with the default setting of either the ATAC-seq/DNase-seq pipeline (available from: https://github.com/kundajelab/atac_dnase_pipelines), or the AQUAS TF and histone ChIP-seq pipeline (available from: https://github.com/kundajelab/chipseq_pipeline), using the human genome GRCh37 as reference. (**STable 2**) Tested sets of CNN hyperparameters. Convolutional neural networks with each set of hyperparameters differing in numbers and sizes of convolutional filters were trained 100 times, for a total of 1000 CNNs trained. (**STable3**) Full list of transcription factor binding motifs with <5% FDR sequence match to motifs activating convolutional filters from the first layers of the 1000 CNN ensemble. No motif redundancy removal was applied here. (**STable 4**) CNN regulatory predictions at 28 T2D association signals fine-mapped to a single most likely causal variant with genetic PPA (gPPA) > = 0.80 or functional PPA (fPPA) > = 0.80. (**STable 5**) CNN regulatory predictions at signals with at least two variants with functional PPAs (fPPAs) > = 0.2. These are the signals where incorporating CNN predictions downstream of fine-mapping can yield the largest benefits. The table lists all the variants with fPPA > = 0.05 at these signals, together with their CNN q-value (lowest_Q), and the corresponding top scoring CNN feature and the mean predicted score difference across the 1000 trained models. (**STable 6**) Primer sequences used for cloning of the Prox1 enhancer. Prox1_enhancer_Forward (Reverse)_internal were designed for sequence validation. Restriction enzymes NheI and XhoI were used for all subsequent cloning. SDM = site directed mutagenesis.

• Transparent reporting form

## Data availability
The datasets analysed during the current study are available in the public repositories under accessions listed in Supplementary file 1-STable 1. The views expressed in this article are those of the author(s) and not necessarily those of the NHS, the NIHR, or the Department of Health.

The following previously published datasets were used:

| Author(s) | Year | Dataset title | Dataset URL | Database and Identifier |
|---|---|---|---|---|
| Matthias Thurner, Martijn van de Bunt, Jason M Torres, Anubha Mahajan, Vibe Nylander, Amanda J Bennett, Kyle J Gaulton, Amy Barrett, Carla Burrows, Christopher G Bell, Robert Lowe, Stephan Beck, Vardhman K Rakyan, Anna L Gloyn, Mark I McCarthy | 2018 | Integration of human pancreatic islet genomic data refines regulatory mechanisms at Type 2 Diabetes susceptibility loci | https://ega-archive.org/studies/EGAS00001002592 | European Genome-Phenome Archive, EGAS00001002592 |
| Ackermann AM, Wang Z, Schug J, Naji A, Kaestner KH | 2015 | Integration of ATAC-seq and RNA-seq Identifies Human Alpha Cell and Beta Cell Signature Genes | https://www.ncbi.nlm.nih.gov/geo/query/acc.cgi?acc=GSE76268 | NCBI Gene Expression Omnibus, GSE76268 |
| Pasquali L, Gaulton KJ, Rodríguez-Seguí SA, Mularoni L, Miguel-Escalada I, Akerman I, Tena JJ, Morán I, Gómez-Marín C, van de Bunt M, Ponsa-Cobas J, Castro N, Nammo T, Cebola I, García-Hurtado J, Maestro MA, Pattou F, Piemonti L, Berney T, Gloyn AL, Ravassard P, Gómez-Skarmeta JL, Müller F, McCarthy MI, Ferrer J | 2014 | Pancreatic islet epigenomics reveals enhancer clusters that are enriched in Type 2 diabetes risk variants | https://www.ebi.ac.uk/arrayexpress/experiments/E-MTAB-1919/ | ArrayExpress, E-MTAB-1919 |
| ENCODE DCC | 2011 | Duke_DnaseSeq_PanIslets | https://www.ncbi.nlm.nih.gov/geo/query/acc.cgi?acc=GSM816660 | NCBI Gene Expression Omnibus, GSM816660 |
| ENCODE DCC | 2012 | UNC_FaireSeq_PanIslets | https://www.ncbi.nlm.nih.gov/geo/query/acc.cgi?acc=GSM864346 | NCBI Gene Expression Omnibus, GSM864346 |
| ENCODE DCC | 2012 | DNaseI/FAIRE/ChIP Synthesis from ENCODE/OpenChrom(Duke/UNC/UTA) | https://www.ncbi.nlm.nih.gov/geo/query/acc.cgi?acc=GSE40833 | NCBI Gene Expression Omnibus, GSE40833 |
| Bhandare R, Schug J, Le Lay J, Fox A, Smirnova O, Liu C, Naji A, Kaestner KH | 2010 | ChIP-Seq of human normal pancreatic islets with anti-histone antibodies to analyse histone modifications | https://www.ebi.ac.uk/arrayexpress/experiments/E-MTAB-189/ | ArrayExpress, E-MTAB-189 |
| Bernstein BE, Meissner A | 2010 | BI Human Reference Epigenome Mapping Project: ChIP-Seq in human subject | https://www.ncbi.nlm.nih.gov/geo/query/acc.cgi?acc=GSE19465 | NCBI Gene Expression Omnibus, GSE19465 |
| Bernstein BE, Stamatoyannopoulos JA, Costello JF, Ren B | 2009 | UCSF-UBC Human Reference Epigenome Mapping Project | https://www.ncbi.nlm.nih.gov/geo/query/acc.cgi?acc=GSE16368 | NCBI Gene Expression Omnibus, GSE16368 |
| Bramswig NC, Everett LJ, Schug J, Dorrell C, Liu C, Luo Y, Streeter PR, Naji A, Grompe M, Kaestner KH | 2013 | Epigenomic plasticity enables human pancreatic alpha to beta cell reprogramming | https://www.ncbi.nlm.nih.gov/geo/query/acc.cgi?acc=GSE50386 | NCBI Gene Expression Omnibus, GSE50386 |
| Stitzel ML, Sethupathy P, Pearson | 2010 | Global epigenomic analysis of primary human pancreatic islets | https://www.ncbi.nlm.nih.gov/geo/query/acc. | NCBI Gene Expression Omnibus, |

| | | |
|---|---|---|
| DS, Chines PS, Song L, Erdos MR, Welch R, Parker SC, Boyle AP, Scott LJ, Margulies EH, Boehnke M, Furey TS, Crawford GE, Collins FS | provides insights into type 2 diabetes susceptibility loci | cgi?acc=GSE23784    GSE23784 |

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
