## [Decision Letter]

**Acceptance summary:**

The reviewers found this to be an exciting application of deep learning CNN models and pancreatic islet epigenomic data to type 2 diabetes (T2D) genome wide association study (GWAS) variants, including the convincing experimental validation of a single-locus effect. Additionally, the bulk enrichment analyses shown in Figure 3 demonstrate the likely broad utility of such an approach. Congratulations on the nice work, and we will look forward to seeing how these models are further used to provide mechanistic insights across diverse complex diseases for which GWAS and tissue-relevant epigenomic annotations are available.

**Decision letter after peer review:**

Thank you for submitting your article "Deep learning models predict regulatory variants in pancreatic islets and refine type 2 diabetes association signals" for consideration by *eLife*. Your article has been reviewed by two peer reviewers, one of whom is a member of our Board of Reviewing Editors, and the evaluation has been overseen by Harry Dietz as the Senior Editor. The reviewers have opted to remain anonymous.

The reviewers have discussed the reviews with one another and the Reviewing Editor has drafted this decision to help you prepare a revised submission.

Summary:

Wesolowska-Andersen et al. present a convolutional neural network (CNN) derived approach to predict regulatory variants in pancreatic islets and refine T2D GWAS signals. Specifically, the authors measure the regulatory potential of T2D associated variants using the CNN trained on diverse islet functional genomics data and compare their predictions to those derived from genetic and functional fine-mapping approaches. They highlight examples of deep learning derived predictions that help them refine signals at multiple loci in a tissue-specific manner. This is a well-written and interesting paper. Perhaps the most striking result is shown in Figure 3 which convincingly shows differential enrichment when T2D GWAS signals are partitioned into insulin action vs. secretion loci. The single locus validation is also quite nice.

Overall, the findings are reasonable and the validation at the end ties the manuscript together nicely. However, we recommend revisions that would improve the overall clarity of the manuscript and associated figures.

Essential revisions:

Showing a cartoon/schematic of the CNN architecture used and how the input and output map to this architecture will be useful for guiding the general readership at *eLife*.

The authors should mention in the Discussion how their prediction method contrasts with doing some experimental rapid high-throughput approach like Hidra.

It is clear that the authors are aiming to make the case that their models are an improvement over previous prediction approaches. Although they make a strong case, one does wonder if pancreatic islets are the best setting in which to initially do this. After all, pancreatic islets represent a mixed cell population, so any epigenomic features will be drawn from all the diverse cells of this tissue. Given there is an abundance of 'pure' cell types with this sort of data available (for example, in ENCODE), it would have utility to 1) run in that setting first to demonstrate the optimal power of this approach for a relevant disease with an abundance of GWAS hits, and then 2) understand what the 'cost' is if you subsequently run in a mixed cell setting like the one they delineate. The problem with this will be with getting a relevant GWAS data set that aligns with a very densely-profiled cell line and other orthogonal validation data. Another alternative "validation" approach could be to compare the models to islet eQTLs, for which there are now good data sets available. The models presented here should be highly enriched for islet eQTL. Our discussion led to the conclusion that implementing one of the above approaches will strengthen the work: either implementing the models in a homogeneous cell line (to circumvent the issues associated with mixed cell populations) or to perform model comparisons with islet eQTL signals.

---

## [Author Response]

Essential revisions:Showing a cartoon/schematic of the CNN architecture used and how the input and output map to this architecture will be useful for guiding the general readership at eLife.

We have now included a cartoon schematic of the CNN architecture used as a new Figure 1—figure supplement 1.

The authors should mention in the Discussion how their prediction method contrasts with doing some experimental rapid high-throughput approach like Hidra.

We added the following paragraph in the Discussion section: “The development of high-throughput experimental methods for assaying regulatory variants, such as massively-parallel reporter assays (MPRAs), offers a complementary route to deriving information on variant-specific function. […] CNN models have the potential to complement high-throughput experimental approaches to elucidate regulatory variants, by learning the regulatory grammar at genomic locations assayed directly in these experiments, and allowing extrapolation to variants (and cell-types) not assayed directly.”

It is clear that the authors are aiming to make the case that their models are an improvement over previous prediction approaches. Although they make a strong case, one does wonder if pancreatic islets are the best setting in which to initially do this. After all, pancreatic islets represent a mixed cell population, so any epigenomic features will be drawn from all the diverse cells of this tissue. Given there is an abundance of 'pure' cell types with this sort of data available (for example, in ENCODE), it would have utility to 1) run in that setting first to demonstrate the optimal power of this approach for a relevant disease with an abundance of GWAS hits, and then 2) understand what the 'cost' is if you subsequently run in a mixed cell setting like the one they delineate. The problem with this will be with getting a relevant GWAS data set that aligns with a very densely-profiled cell line and other orthogonal validation data. Another alternative "validation" approach could be to compare the models to islet eQTLs, for which there are now good data sets available. The models presented here should be highly enriched for islet eQTL. Our discussion led to the conclusion that implementing one of the above approaches will strengthen the work: either implementing the models in a homogeneous cell line (to circumvent the issues associated with mixed cell populations) or to perform model comparisons with islet eQTL signals.

We fully acknowledge reviewers comment that pancreatic islets are indeed a mixture of cell types, and while function of the insulin-producing beta cells is imperative to blood glucose control and T2D aetiology, there is emerging evidence that other cell types may contribute as well. We would also like to bring to the reviewers’ attention that among the 30 chromatin features included in our networks inputs and predictions, there were several epigenomic datasets measured in FACS-sorted pure islet cell populations – specifically beta cells, alpha cells, as well as acinar and exocrine cells. We did not see many differences between these pure cell population predictions, possibly because the open chromatin and the H3K4me3 profiles for these cell populations represented pairs of features with highest pairwise similarity, as shown in the feature similarity plot using Jaccard metric.

**Author response image 1. respfig1:** Pairwise Jaccard distances for the pancreatic islet epigenomic datasets used in CNN training.

Given that, we believe that when looking for cell-type specific differences in epigenomic profiles, it would be important to take into consideration the quantitative differences in open chromatin, or levels of chromatin modifications, rather than investigate them as binary traits.

As the reviewer rightly suggests, to comprehensively test this hypothesis, we would require good quality data for several pure cell types, or cell lines, as well as an appropriate set of GWAS results, which is not trivial. Additionally, this would require setting up a completely new scheme for CNN training, which due to computational time, would not be compatible with the two month turnaround time for the manuscript. We address this concern in Discussion: “Additionally, even within the correct tissue, inference can be complicated by the fact that tissues represent heterogeneous mixtures of cell populations, of which only a subset might be directly relevant to disease aetiology. The growing availability of cell-type specific functional genomic datasets will make it possible to explore the extent to which this more precise assignment of disease pathology allows more accurate inference regarding the causal mechanisms.”

We took forward the reviewer’s suggestion for the additional analysis comparing the CNN results to pancreatic islet eQTL data. Initially, we did not include such comparison in our manuscript as previous islet eQTL studies discovered that the overlap between T2D GWAS signals and islet eQTLs does only explain a small proportion of the GWAS signals, even though there exists a significant enrichment of GWAS hits within the annotated eQTLs.

In our manuscript, we have compared our CNN predictions at the credible set variants the recent human pancreatic islet eQTL study in 420 islet donors from the INSPIRE consortium (Viñuela et al., 2019). The results are summarized in a new Figure 2B and new paragraph in the Results section: “To further validate the functional inference from the islet CNN, we compared the CNN predictions at the credible set variants with results from a recent cis-eQTL study of human islet samples from 420 donors performed by the InsPIRE consortium (Viñuela et al., 2019). […] We considered the enrichments of variants predicted to affect the six different groups of chromatin features, and found that variants predicted to affect promoter or enhancer activities in islets were preferentially enriched among the top pancreatic islet eQTL results (Figure 2B).”